# Double Crack Damage Identification of Welded Steel Structure Based on LAMB WAVES of S0 Mode

**DOI:** 10.3390/ma12111800

**Published:** 2019-06-03

**Authors:** Muping Hu, Xiaodan Sun, Jian He, Yangyang Zhan

**Affiliations:** College of Aerospace and Civil Engineering, Harbin Engineering University, Harbin 150001, China; 2013021113@hrbeu.edu.cn (M.H.); sunxiaodan@hrbeu.edu.cn (X.S.); 2013021325@hrbeu.edu.cn (Y.Z.)

**Keywords:** Lamb waves, double structural damage identification, weld damage identification, ellipse localization method, numerical simulation

## Abstract

Steel structures are widely used in large-span bridges, offshore platforms, mining equipment and other large-scale buildings. The damage of steel structures will cause significant safety risks in a project. Therefore, it is of great significance to identify and study damage to steel structures. In this study, the propagation of Lamb waves in a steel plate with double cracks is simulated. Using finite element analysis and experimental study, damage identification and damage imaging of double crack damage to a steel plate are performed, and the numerical simulation results are in good agreement with the experimental results. Considering the reflection and transmission of Lamb waves at the welding seam, the location and imaging of crack damage in a welded steel plate are also studied. The imaging results obtained from simulation and experiment show high level in accuracy. By comparing the amplitude of the signal in the propagation process, it is concluded that the transmission energy at the weld seam decreases.

## 1. Introduction 

With the continuous development of science and technology, nondestructive detecting technology has been widely used in various fields. In the field of nondestructive assessment and structural health monitoring, the most common damage detection techniques include visual inspection, optical fibers, shear force photography, infrared thermal imaging, eddy current, Lamb waves, and mechanical impedance. For example, in the field of damage identification using acoustic emission (AE), by applying a source localization algorithm which uses the registered acoustic emission signals from the sensor networks, Sikdar et al. [1] identified the damage source locations efficiently. In order to tackle uncertainties of damage detection in composite structures within a framework of automated probabilistic damage detection, some scholars used experimental data to train a parameterized input–output probability model of AE system and gave the probability description method of damage location [2]. An acoustic emission (AE) based damage detection method, combined with real-time health monitoring framework to identify the probable damage initiation locations in advanced sandwich composite structures was proposed [3]. Amongst damage detection methods, Lamb waves [4,5] and electromagnetic interference [6] have been increasingly applied in recent years because of their high sensitivity to structural damage.

In 1917, the equations of Lamb waves were discovered by Lamb [7]. Since then, many scholars have studied the characteristics of Lamb waves and have demonstrated its applicability in the field of nondestructive detection. Lamb waves have been an important means of nondestructive testing. Through some experimental and theoretical investigations, Lamb waves have been proved to be suitable for locating an approximate sizing of structural damages [8]. He et al. [9] used the finite element method to simulate the propagation of Lamb waves. The location of crack damage is simulated by the amplitude addition method. It has been proved that through research on the propagation of Lamb waves, that when there is damage in the sheet, S0 and A0 will produce new wave packets by mode conversion [10]. Based on the propagation characteristics and wave theory of thin plates, it can be proven that Lamb waves have multimode characteristics which are related to the frequency–thickness product [11]. 

Through systematic analysis of the dispersion of Lamb waves in aluminum structures, Li et al. [12] differentiated the mode of each package in the signals of Lamb waves and raised an effective approach for damage localization. A systematic method for dispersion analysis of Lamb waves (S0 and A0 waves) using wavelet transform was proposed by Hosseinabadi et al. [13] and they applied guided ultrasonic waves for damage detection in structural health monitoring. Some scholars used the change of flight time and the phase of the Lamb waves to detect damage. Using different signal processing technologies such as Fourier and Hilbert transforms, the quality of the guided wave signal was improved when identifying damage size [14].

Based on these studies of Lamb waves’ characteristics, nondestructive testing technology has also been greatly developed. With the continuous development of science and technology, nondestructive detecting technology has been widely used in various fields. As nondestructive testing techniques are viewed as viable solutions to meet requirements like high testing efficiency, high reliability, real-time, visualized and environmental friendly, Zeng et al. [15] applied this technology to the damage detecting of composite structure, and proposed a damage imaging method that accommodates for multipath scattering Lamb wave signals. Through modeling and experiments, Takpara et al. [16] validated the optimization of parameters for the interdigital transducer (IDT) sensor with piezoelectric ceramics transducer (PZT). Nowadays, metal materials play an important role in many domains. However, different kinds of metal defects, such as cracks and contraction cavities will directly affect the metal service life and mechanical properties. Numerous nondestructive testing (NDT) methods have been proposed for detecting metal defects. A nondestructive detecting method for metal material defects based on multimodal signals is proposed to expand the scope of detection and obtain more complete information [17]. Muskar and Yelve [18] determined structural damage by imaging technology and proposed Euclidean and Lagrangian optimization of experimental data, which further refined the damage location in the closed area. In addition to damage localization, damage imaging can also be achieved. Relying on both experimentally collected natural frequencies and frequencies calculated using a mathematical model, a method for detecting cracks was presented, which showed exceptional convergence on the size and location of cracks [19].

The above-mentioned scholars not only studied the dispersion curve and propagation characteristics of Lamb waves but also studied the damage imaging and damage location on the basis of considering Lamb characteristics. These studies have promoted the development of nondestructive testing technology. However, in practical engineering applications, the damages are often more complex. There are fewer cases of single damage and most of the works from the foregoing scholars focus on the identification of single damage. Therefore, this paper studied the double damage localization and imaging of steel plates based on the symmetric Lamb waves excitation. Numerical simulation and experiments of Lamb waves propagation in a steel plate were carried out, and damage imaging results consistent with the actual situation were obtained. It is proved that the active Lamb waves acoustic emission method is not only suitable for single damage monitoring but also for multi-damage monitoring. Also, this paper studied the damage imaging in welded steel plates and analyzed the influence of the welded joint on signal amplitude attenuation, which both have great guiding significance to the engineering application of NDT on welded plates.

## 2. Basic Theory of Lamb Waves

### 2.1. Basic Theory of Group Velocity and Phase Velocity

In the dispersion of Lamb waves, symmetric as well as anti-symmetric modes have two important parameters: group velocity and phase velocity. When Lamb waves propagate in a plate, the phenomenon of dispersion will occur gradually, and the waves will propagate at different modes and velocities. In each dispersion wave, each independent wave still propagates at its own speed, which is the corresponding phase velocity. When waves propagate in a plate, in-phase superposition occurs, which changes the observed amplitude. Therefore, when analyzing the superposition of two waves with the same amplitude and different frequency, the following formula can be obtained: (assuming that the frequencies of the two waves are *ω*_1_ and *ω*_2_).
(1)$$f(x,t)=H\cos(k_{1}x-\omega_{1}t)+H\cos(k_{2}x-\omega_{2}t)$$
where *k*_1_ = *ω*/*c*_1_, *k*_2_
*=*
*ω/c*_2_. *k*_1_, *k*_2_ are wave numbers; *H* is amplitude. The formula can be deduced as:(2)$$f(x,t)=2H\cos(\frac{k_{1}-k_{2}}{2}x-\frac{\omega_{1}-\omega_{2}}{2}t)\cos(\frac{k_{1}+k_{2}}{2}x-\frac{\omega_{1}+\omega_{2}}{2}t)$$

Suppose:(3)$$(k_{1}+k_{2})/2=k_{n}$$
(4)$$(k_{1}-k_{2})/2=\Delta k$$
(5)$$(\omega_{1}+\omega_{2})/2=\omega_{n}$$
(6)$$(\omega_{1}-\omega_{2})/2=\Delta \omega$$
then, the following is obtained:(7)$$f(x,t)=2H\cos(\Delta kx-\Delta \omega t)\cos(k_{n}x-\omega_{n}t)$$

Then, two waveforms of different frequencies are superposed, that is, the superposition of a high-frequency wave and low-frequency wave; the high-frequency propagation velocity is defined as phase velocity *c_p_ =*
*ω_c_/k_c_*. Simultaneously, low frequency is defined as group velocity *c_g_ =* Δ*ω*/Δ*k*, and it is also *c_g_ = d**ω/dk*. By defining group velocity as phase velocity, the formula for calculating group velocity can be obtained as follows:(8)$$c_{g}=\frac{d\omega }{dk}=\frac{c_{p}}{1-\frac{\omega }{k}\frac{dc_{p}}{d\omega }}$$

### 2.2. Dispersion Curve

As the formulas for solving group velocity by phase velocity have been given above, the above equations can be solved numerically by using MATLAB, and the frequency equation can be solved according to the corresponding relationship between group velocity and phase velocity. When programming, the frequency–thickness products are related to velocity. The material was Q235 steel plate with 4-mm thickness. The parameters are as listed in Table 1, where E is the elastic modulus of the steel plate, *v* is Poisson’s ratio, *ρ* is the density of the material, d is the thickness of the steel plate, c_L_ is the velocity of the longitudinal wave, c_T_ is the velocity of the transverse wave. The output result is shown in Figure 1. In the figure, the frequency is set to the abscissa and the velocity to the ordinate. From the figure, we can see that there was a dispersion effect in the Lamb waves, and there were many modes. Under the excitation of different frequencies, the same plate was subjected to different situations.

## 3. Identification and Imaging of Double Crack Damage Based on the Lamb Waves of the S0 Mode

### 3.1. Geometric Model for Numerical Simulation

In practical engineering, the damage state is usually very complex and may have double or more cracks at the same time. Therefore, this paper further investigates the damage identification method of plate structures when double damage occurs. The material parameters of the steel plate are listed in Table 1.

The numerical modelling was performed by using ABAQUS® ver. 6.14 finite element code, as shown in the Figure 2, the size of steel plate was 800 mm × 800 mm × 4 mm, and the coordinate points (−242.5,0) and (205,0) were taken as the crack damage centers. The crack with a length of 80 mm and a width of 5 mm was set in the left side, and the crack with a length of 40 mm and a width of 5 mm was set in the right side, and the damage of the plate was a rectangular area passing through the thickness direction. The position coordinates of the transducers are listed in Table 2. As shown in Figure 3, there were 18 sensors in total, PZT1–PZT9 are arranged on the top of the plate, PZT1’–PZT9’ are arranged on the bottom of the plate, PZT1–PZT9 are symmetrically distributed with PZT1’–PZT9’ along the direction of plate thickness.

### 3.2. Excitation Signal

To simulate active acoustic emission by the finite element method, the appropriate excitation signal must be determined first. Compared with broadband signals, narrowband signals make it easier to distinguish each wave packet, and wave packets do not overlap easily. Narrowband sinusoidal signals are usually used as the excitation waveform. Therefore, the narrowband sinusoidal wave function modulated by the Hanning window was adopted in this study. The expression of the narrow-band sinusoidal wave function is as follows:(9)$$y=0.5\times \left[1-\cos(\frac{2\times 2\pi \times f\times t}{n})\right]\times \sin(2\times 2\pi \times f\times t)$$
where: *f*—frequency of excitation signal, Hz; *n*—periodic number of excited signals.

There are three excitation modes of the Lamb waves: unilateral excitation, symmetric excitation, and anti-symmetric excitation, which are as shown in Figure 4. Unilateral excitation was not conducive to signal extraction because it generated two different modes of excitation wave. Symmetric excitation generated an excitation wave of a single S0 mode, anti-symmetric excitation generated excitation wave of single A0 mode. The wave group of S0 mode was faster, that was conducive to wave packet separation. Therefore, the symmetric excitation mode was used to excite the Lamb waves. 

Considering the mode of Lamb waves generated in the plate under low-frequency excitation was simpler, the excitation frequency of 147 kHz was selected in the numerical simulation and experiment. The 3.5-period sinusoidal wave had more obvious characteristics than other types of waves: it had five wave peaks with symmetrical distribution, obvious differences between the upper and lower waveforms, and a different number. In addition, the time-domain width of this kind of wave is narrower, which was more conducive to the experiment. Therefore, the excitation signal was designed to be a sinusoidal wave with a period of 3.5 modulated by a Hanning window, which can be seen in Figure 5.

In this model, the excitation was demonstrated as a tangentially concentrated load, the load was a time period dependent dynamic load. The model was a free plate with four sides, the direction of the concentrated load was the tangential direction. This study established a three-dimensional model directly and simulated the establishment of cylindrical coordinates at the excitation load at the place of the transducers as shown in Figure 6.

### 3.3. Mesh Size

The mesh was divided by hexahedron structure technology. The material parameters of the steel plate are given in Table 1, and the mesh length can be calculated based on the data in the table. The calculation process of the mesh length is as follows, in which the velocity of the shear wave is calculated as:(10)$$c_{T}=\sqrt{\frac{\mu }{\rho }}=\sqrt{\frac{G}{\rho }}=\sqrt{\frac{76.9\times 10^{9}\;N/m^{2}}{7850\;kg/m^{3}}}=3130.35\;m/s$$

The simulated frequency is 147 kHz and the minimum wavelength (*λ*_min_) can be calculated as:(11)$$\lambda_{\min}=\frac{c_{T}}{f_{\max}}=\frac{3130.35\;m/s}{147\times 10^{3}\;s^{-1}}=0.0212948\;m=21.3\;\mathrm{mm}$$

The best principle of mesh generation is to transfer the minimum wavelength to 10 units [20]. To satisfy this principle, the mesh length is calculated as follows:(12)$$L_{e}\leq \frac{\lambda_{\min}}{10}=2.13\;\mathrm{mm}$$

To facilitate the excitation of signals and the calculation of simulation results, it was best to take the number of meshes as an integer in thickness and length because the thickness of the steel plate was 4 mm; in the finite element model, the length of mesh elements can be set to 2 mm, 1 mm and 0.8 mm. In theory, the higher the density of the grid, the higher the accuracy of the results. However, overly dense grids will greatly increase the calculating time, so we should select the appropriate size of the grid within the acceptable accuracy range. Therefore, in the finite element model in this study, 1 mm was chosen as the mesh size. As a result, the plate was modelled with 2,560,596 three-dimensional eight-node linear brick elements (C3D8R). 

### 3.4. Elliptical Location Method

When studying the propagation of Lamb waves in the steel plate with double damages, as shown in Figure 7, PZT1 was set as the actuator and PZT2 as the receiver. There were four paths for Lamb waves’ propagation in the plate, as follows: (1) The direct wave from the actuator to the receiver traveled along with the straight line d; (2) Transmitted from the actuator, the wave reached the receiver after the first damage reflection, and its propagation path was r_1_ + r_2_; (3) Transmitted from the actuator, the wave reached the receiver after the second damage reflection, and its propagation path was r_3_ + r_4_; (4) Transmitted by the actuator, the signal first reached damage 1, then reached damage 2, and finally reflected to the receiver. Thus, the path obtained after multiple reflections was r_1_ + s + r_4_.

When analyzing double damage cases, the same method of single damage identification is often used to analyze the path of double damage cases, which is referred to as the elliptical positioning theory. That is, the damage location is located on an elliptical arc, and the driver and the receiver are on the foci of the ellipse respectively. Thus, a complete elliptical model can be established. However, compared to the single damage model, the double damage model has more flight time and time delay. Simultaneously, we know that a path can form two different elliptic equations corresponding to it. When there are enough elliptic equations, the damage can be identified and located. If the elliptic characteristic parameters are obtained, then the following equations can be used to solve the problem:(13)$$\{\begin{array}{c} \frac{r_{1}}{v_{0}}+\frac{r_{2}}{v_{1}}-\frac{d}{v_{0}}=\Delta t_{1}\\ \frac{r_{3}}{v_{0}}+\frac{r_{4}}{v_{1}}-\frac{d}{v_{0}}=\Delta t_{2} \end{array}$$

### 3.5. Simulation Results

The information contained in the received signal of the Lamb waves in the plate with double damages will become more complex. It was found that besides the Lamb waves propagating linearly from the actuator to the receiver, there were also waves carrying the damage signal reflected by damage-one and damage-two in the plate, and there were some reflected waves that reached the plate boundary through different paths. When the waves in the board overlap with each other, the information becomes more complex. Similarly, in the process of solving time parameters, compared with single damage state, the data complexity of the double damage situation makes it more difficult to solve and collect. It can be seen from this that when the double damage condition was analyzed, the Lamb waves of S0 single mode could be used to reduce the interference of the above complex information. In the case of testing, one group of PZTs was selected as the actuators, while the remaining groups were used as the devices receiving signals. As symmetrical excitation was used, the Lamb waves of single S0 mode should have been received at last. Figure 8 shows the propagation stress diagram of Lamb waves with excitation time ranging from 2.5 × 10^−5^ s to 5.0 × 10^−5^ s, and with a time interval of 5 × 10^−6^ s.

The symmetrical excitation signal was applied at PZT5, and the received signal at PZT8 was taken as an example. The signals of Lamb waves of S0 mode were collected from the intact steel plate model and damage model as shown in Figure 9.

By comparing the two figures, in the intact steel plate and the double-damage steel plate, the excitation and reception sensors were at the same position but they had different signals. The two additional wave packets in Figure 9b are the reflection wave packets of the excitation signal passing through damage 1 and damage 2. The first wave packet, which is transmitted directly from PZT5 to PZT8, remains unchanged and is not affected by cracks. MATLAB software (2017a, The MathWorks, Natick, MA, USA) can be used to extract the arrival time of the two damage wave packets.

Nine PZTs were used to excite signals, and the other eight PZTs were used to receive signals. The waveforms were judged and analyzed, and 72 sets of data were obtained for each crack. Based on the arrival time of damage wave packet and ellipse location technology, the ellipse equation could be solved using MATLAB, and the graph could be plotted to locate the damage. Because 72 sets of data can be obtained from the table, 72 ellipses can be drawn by MATLAB. The intersection of these ellipses is the damage location. The damage could be imaged by using the 2D nuclear density map of ORIGIN software (2017, Northampton, MA, USA). The image of the 80 mm crack damage on the left side and the 40 mm crack damage on the right side are shown in Figure 10, both were in good agreement with the actual crack location.

It can be seen from the figure that when there are two cracks in the plate, the multi-damage signals are complex and interfere with each other, and only if the signals are processed to confirm the arrival time of the damage wave packet, according to the damage location and imaging technology, the imaging results can be in good agreement with the actual situation, which verifies the effectiveness of this method in the multi-damage steel plate.

### 3.6. Experiment on the Identification of Double Crack Damage

The entire experiment was carried out on the isolation table, and Lamb waves were excited in the specimen by an arbitrary waveform generator (AFG31052, Tektronix, Hong Kong, China). The excitation wave was output by two lines, one of which was input into the first channel of the oscilloscope (MSO545-BW-350, Tektronix, Hong Kong, China), and the other one was input into the experimental plate after being amplified by the signal amplifier. Other channels of the oscilloscope received the signals from receiving sensors. By analyzing the signals in the oscilloscope, we studied the propagation rule of Lamb waves in the plate and analyzed whether there was damage in the structure. The complete experimental process is shown in Figure 11.

Two cracks were set in the complete steel plate. The cracks and PZTs layout are shown in Figure 12. The type of PZT was YT-5L (Beijing railway scientific instrument equipment Co. Ltd., Beijing, China) with a diameter of 10mm and the thickness of 1 mm. The coordinate system can be seen in the figure, the central positions of the two cracks were (−245 mm, 0) and (205 mm, −60 mm) respectively. The length of crack 1 was 80 mm and the width was 5 mm, the length of crack 2 was 40 mm and the width was 5 mm.

The signals in the experiment were relatively complex. To analyze the arrival time of the damaged wave packet, in addition to the subtraction signal processing, Hilbert transformation was also required, which allowed for the arrival time of the damaged wave packet to be accurately obtained. Imaging results are shown in Figure 13 below. It can be found that the damage imaging results of the left and right cracks were in good agreement with the actual crack, which verifies the accuracy of this method in the experiment.

## 4. Damage Identification of Welded Steel Plate Based on S0 Mode

### 4.1. Numerical Simulation

As shown in Figure 14, the structural geometric model was formed by welding two Q235 steel plates with dimensions of 600 mm × 300 mm × 4 mm. The damping factor of the material was set at 0.005. The parameters of the material can be seen in Table 1 in Section 2.2. On the left side, 10 PZTs were arranged symmetrically on both sides of the steel plates, and the serial numbers were PZT1–PZT5 and PZT1’–PZT5’. Five PZTs were arranged on one side of the right steel plate, and the serial numbers were PZT6–PZT10. The coordinate system of the left steel plate is x-y, the coordinate system of the right steel plate is x’-y’, as shown in the Figure 14. The specific coordinates of the PZTs are listed in Table 3.

In practical engineering, cracks may occur not only in the center of the steel plate but also in the weld of the welded steel plate. Therefore, a crack with a length of 40 mm and a width of 5 mm was applied at the joint of the left steel plate. Among them, in order to meet the convenience of integer grid, an arc with a radius of 2 mm was adopted for welding parts, hexahedral structure technology was adopted for grid division. The grid diagram of welded steel plate is shown in Figure 15. The chosen mesh size was 1 mm in the simulation model. As a result, the left plate was modeled with 717,220 three-dimensional eight-node linear brick elements (C3D8R), the right plate was modeled with 713,376 C3D8R, and the weld joint was modelled with 3000 C3D8R.

### 4.2. Numerical Simulation Results

Symmetrical excitation signals were applied with PZT3 and PZT3’ as a group. Figure 16 shows the propagation of Lamb waves in welded steel plates from 2.5 × 10^−5^ s to 5.0 × 10^−5^ s with a time interval of 5 × 10^−6^ s.

As seen from Figure 16, Lamb wave signals will be reflected upon encountering welding seams and cracks. As the crack was at the weld, the arrival times of the reflection wave generated by the weld and the reflection wave generated by the crack were very close. For example, when the excitation was applied at PZT5–PZT5’, the collected signals received by PZT3 and PZT10 are shown in Figure 17 and Figure 18.

As seen from the Figure 18, when the crack was at the weld, the amplitude of the received signal of PZT10 significantly decreased, and more signals were reflected at the weld and the crack. Because the crack was at the welding seam, the receiving signal of PZT3 between the complete plate and the damaged plate were not much different, so the reflected wave packet at the crack could not be directly observed. Therefore, the differences between the signals obtained from the welded steel plates with cracks and those without cracks were calculated.

Based on the damage signal, the flight time of the Lamb waves at the crack caould be calculated. Using elliptic positioning technology, the damage could be located by solving the elliptic equation with MATLAB and by plotting a graph, and the damage could be imaged by using ORIGIN software core density map with full amplitude addition. The imaging results are shown in Figure 19 and were in good agreement with the actual location of cracks. Although the appearance of welding seams had an impact on the propagation of Lamb waves, it did not affect the application of damage location and imaging technology.

Attenuation analysis of signal energy can be performed as follows: by comparing Figure 17a with Figure 18a and Figure 17b with Figure 18b, it can be seen that signal energy decreased with the increment of distance regardless of whether there were damages in structural components. Another reason PZT10 had less energy than PZT3 was that part of the reflection occured at the weld. By comparing Figure 17a with Figure 17b and Figure 18a with Figure 18b, it can be seen that when there was a damage in structural components, most Lamb waves were reflected when they reached the damage, which considerably reduced the amplitude and energy of the signal received by PZT10.

Therefore, the following two conclusions can be drawn: (1) the signal energy decreased with the increment of distance; (2) both the welding seam and the damage reflected the signal, making the signal energy of the propagation path through the welding seam and the damage decrease.

### 4.3. Experiment

#### 4.3.1. Experimental Results

As presented in Table 2 and Table 3, the piezoelectric ceramic sensors were symmetrically pasted on the welded steel plate on both sides, and the cracks were set above the weld 40 mm long and 5 mm wide, as shown in Figure 20.

Experiments were carried out on the welded intact steel plate and the damaged steel plate, respectively, to obtain the signal. The signals were denoised, Hilbert transformed and subtraction processed to obtain the arrival time of the damage wave packet. Then, the time delay and other information were calculated according to the data, and the damage was positioned according to the elliptic equation. The imaging results are shown in Figure 21. It can be found that the damage imaging results were in good agreement with the actual crack, which verifies the accuracy of this method in the experiment.

#### 4.3.2. Attenuation Analysis of Signal Energy 

Lamb waves propagated in the steel plate, and their amplitude gradually decreased with the increase of distance; in the welded steel plate, the amplitude decreased more dramatically with the reflection at the welding seam. With 147 kHz as the central frequency, in the experiment, the distance between each receiving point and the excitation point and the voltage of the received signals were measured, as listed in Table 4. In the numerical simulation, the distance between each receiving point and the excitation point and the displacement of the received signals were measured, as listed in Table 5. The comparison between numerical simulation results and experimental results is shown in the Figure 22. Since the energy received by the sensor on the right plate was too low compared to that received by the sensor on the left plate, in order to show clearly the attenuation of the signal energy, the logarithmic coordinate was applied in the ordinate.

As can be seen from the figure, the experimental results and numerical simulation results were in good agreement. The amplitude of the Lamb waves decreased with the increase of distance. Therefore, the energy of S0 mode also decreased with the increment of the distance, and the amplitude of Lamb waves passing through the welding seam to the other side of the plate decreased obviously; it can be seen that a large number of waves will be reflected at the weld. The following can be concluded: 1. the signal energy decreased with the increase of distance; 2. that welding seam and the damage could reflect the signals making the energy of the signals propagating in the paths, which needed to pass through the welding seam and the damage, decrease significantly. The attenuation law of signal energy obtained in the experimental study was the same as that obtained in the numerical simulation.

## 5. Conclusions

In this paper, the equation of Lamb waves was derived on the basis of the relevant theories of elasticity, and the propagation of Lamb waves in steel plates was studied. Based on the theory and simulation, the structure was studied experimentally and damage identification was carried out. The main conclusions are as follows:The damage imaging results of the steel plate with double cracks from numerical simulation and experiments were in good agreement. The damage location shows a high level of accuracy. The mutual verification of finite element method and experiments proved the reliability of the Lamb waves monitoring method.When reflected by damages, Lamb waves had amplitude attenuation, which could be reflected in the echo signal. As can be seen from the imaging results of double damages, the differences in the damage length were quite obvious. It was proved that the method is sensitive to the length of damage. This will contribute to research on the length of the structures’ crack in further study. When studying damage location in welded steel plates, it is concluded that a part of the Lamb waves would reflect at the weld and would partly pass through the weld. The results of numerical simulation and experiments confirm this inference. The agreement of simulation results and experimental results prove the feasibility of the application of the Lamb wave method in the welded structure.By analysing the amplitude and distance of the signal in the welded steel plate, it is concluded that the energy of Lamb wave in the steel plate decreased with the increment of the distance. The welding seam reflected most Lamb waves, and the energy considerably reduced when they passed through the welding seam during the propagation process.

## Figures and Tables

**Figure 1 materials-12-01800-f001:**
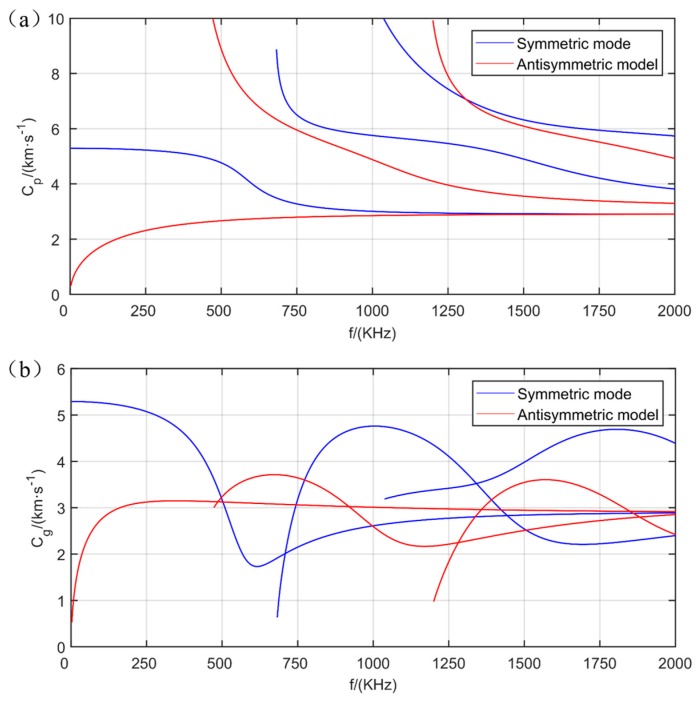
Dispersion curve of plate: (**a**) phase velocity dispersion curve of 4-mm steel plate, (**b**) group velocity dispersion curve of 4-mm steel plate.

**Figure 2 materials-12-01800-f002:**
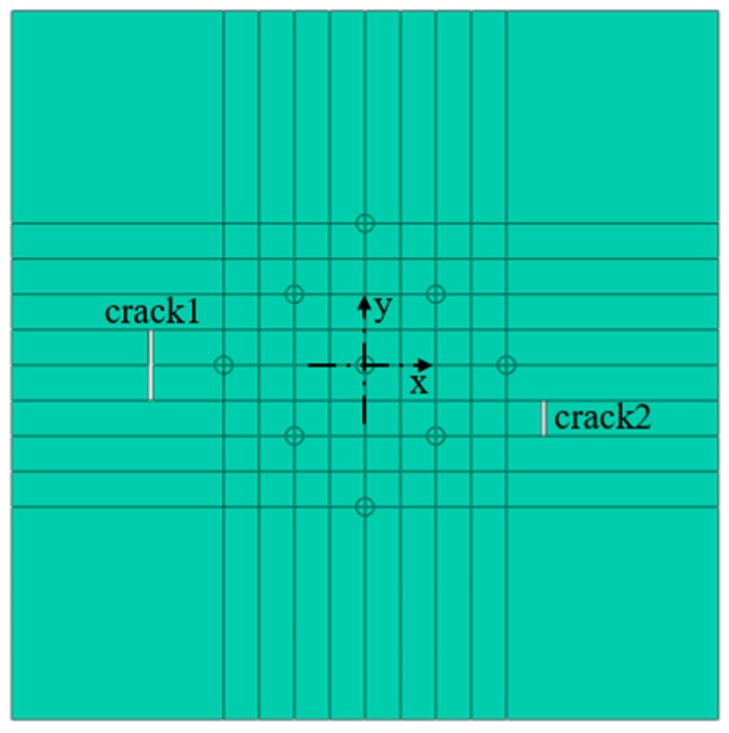
Top view of the geometrical model of steel plate.

**Figure 3 materials-12-01800-f003:**
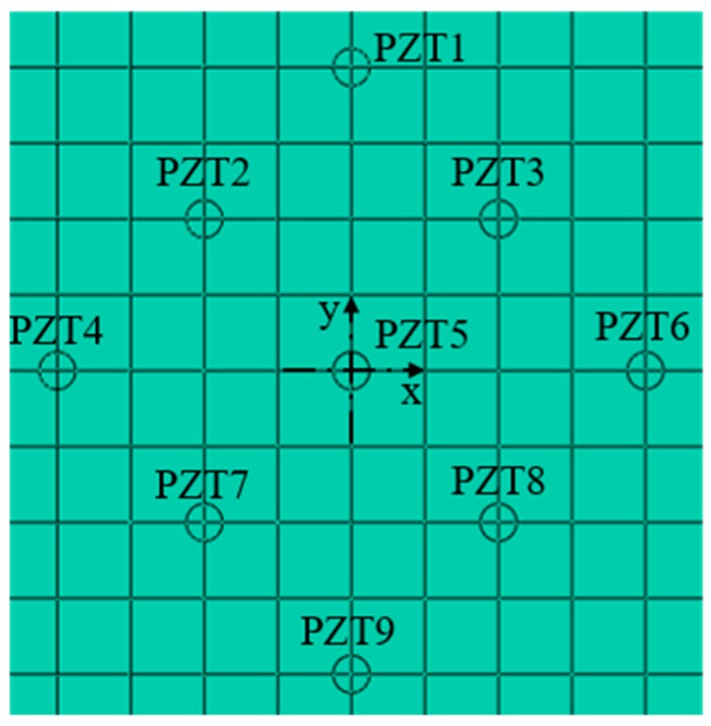
Nomenclature of the piezoelectric ceramics transducers (PZTs).

**Figure 4 materials-12-01800-f004:**
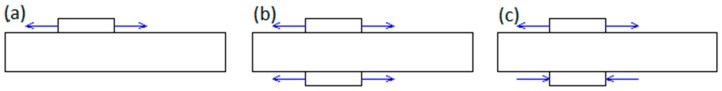
Excitation modes of the Lamb wave: (**a**) unilateral excitation, (**b**) symmetric excitation, (**c**) anti-symmetric excitation.

**Figure 5 materials-12-01800-f005:**
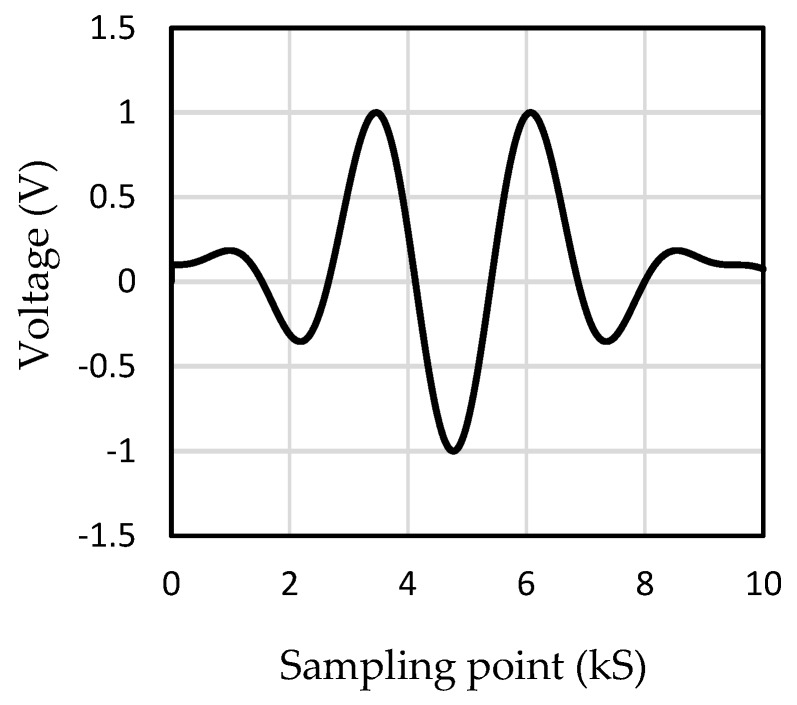
Sinusoidal wave modulated by a Hanning window (kS: kilo sampling points).

**Figure 6 materials-12-01800-f006:**
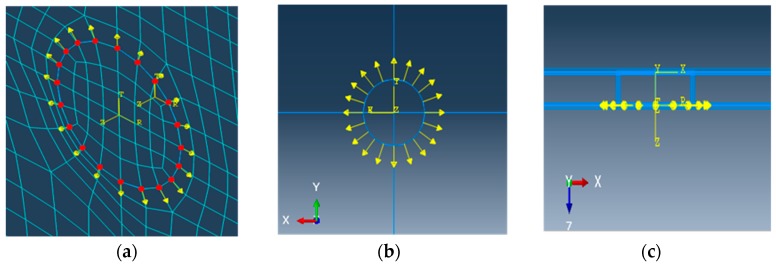
Load simulation in ABAQUS: (**a**) schematic diagram of excitation point, (**b**) top view of the excitation load, (**c**) side view of the excitation load.

**Figure 7 materials-12-01800-f007:**
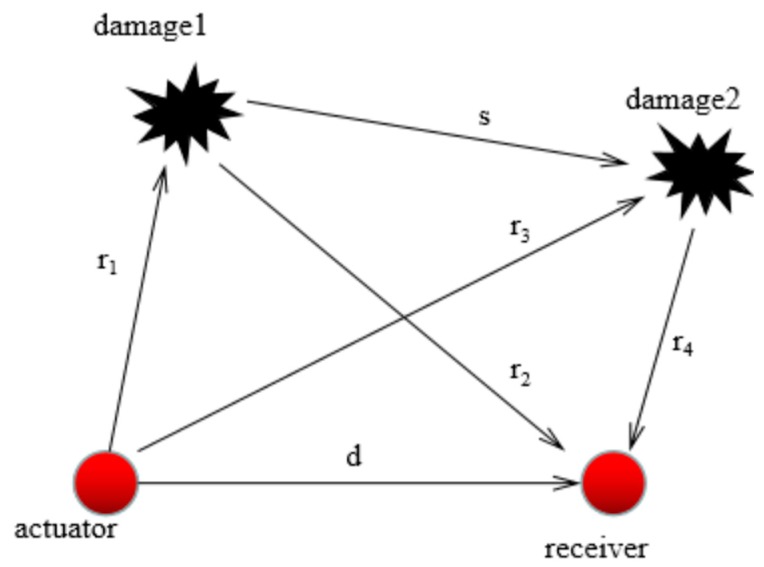
Propagation of Lamb waves under dual damage.

**Figure 8 materials-12-01800-f008:**
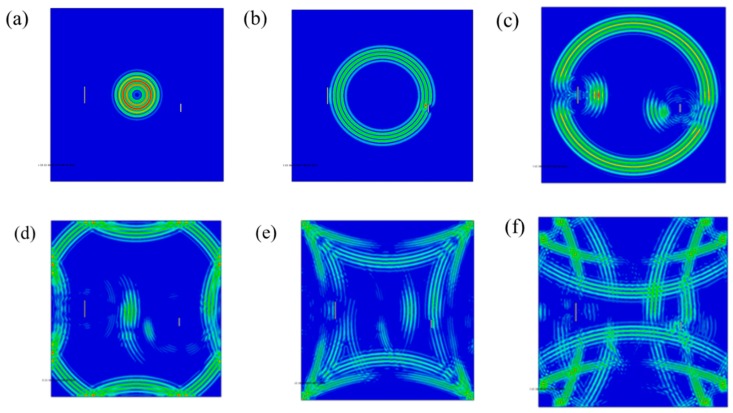
Schema of Lamb waves propagation in a double damaged steel plate: (**a**) 2.5 × 10^−5^ s, (**b**) 3.0 × 10^−5^ s, (**c**) 3.5 × 10^−5^ s, (**d**) 4.0 × 10^−5^ s, (**e**) 4.5 × 10^−5^ s, (**f**) 5.0 × 10^−5^ s.

**Figure 9 materials-12-01800-f009:**
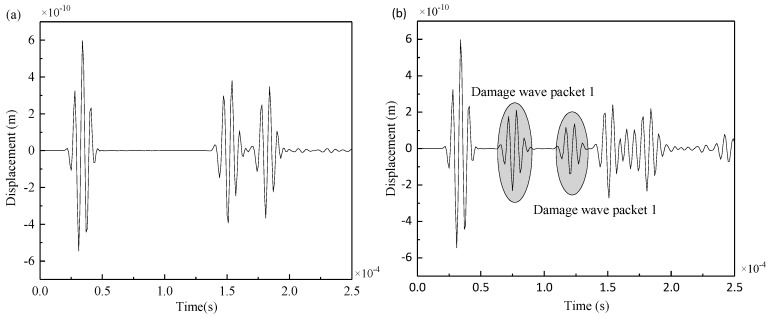
(**a**) Received signal at PZT4 of intact steel plate, (**b**) received signal at PZT4 of damaged steel plate.

**Figure 10 materials-12-01800-f010:**
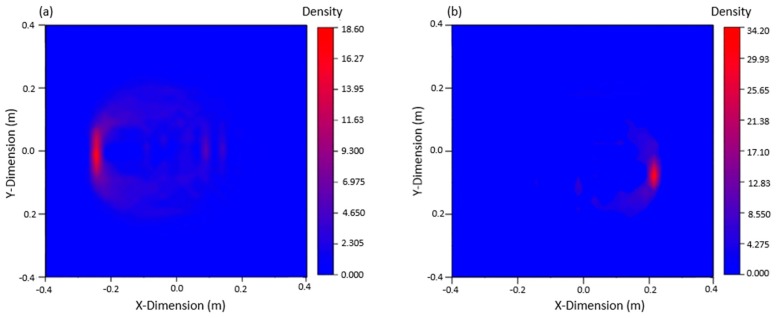
Damage imaging: (**a**) damage imaging of the left 80 mm crack, (**b**) damage imaging of the right 40 mm crack.

**Figure 11 materials-12-01800-f011:**
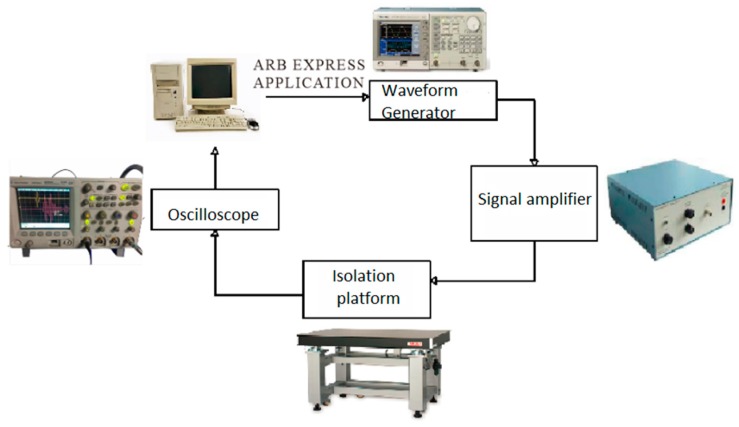
Experimental process.

**Figure 12 materials-12-01800-f012:**
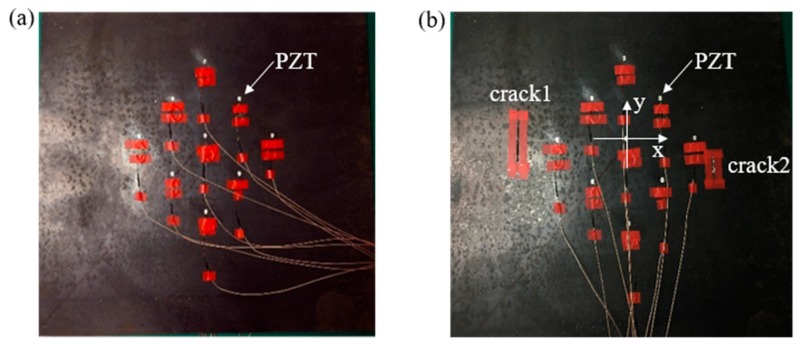
The damage and PZT layout in experiments: (**a**) intact steel plate, (**b**) damaged steel plate.

**Figure 13 materials-12-01800-f013:**
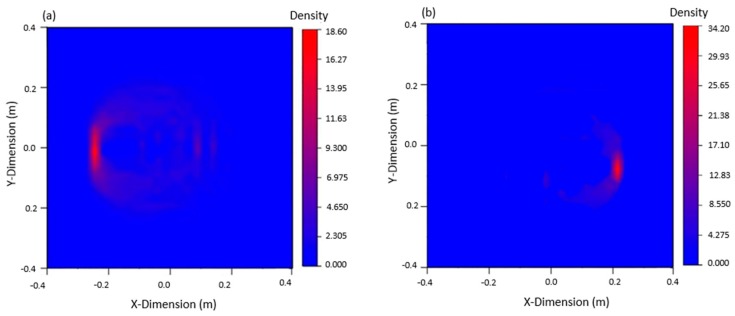
Damage imaging: (**a**) damage imaging of the left 80 mm crack, (**b**) damage imaging of the right 40 mm crack.

**Figure 14 materials-12-01800-f014:**
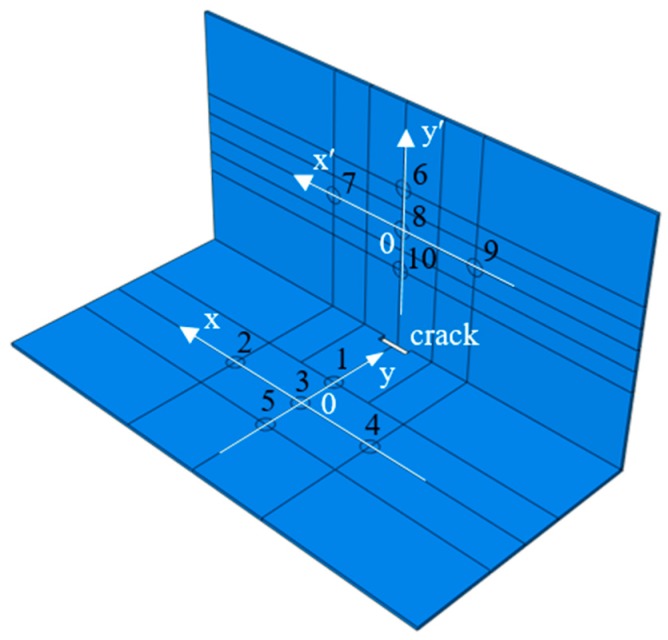
Geometric model and the layout of the PZTs.

**Figure 15 materials-12-01800-f015:**
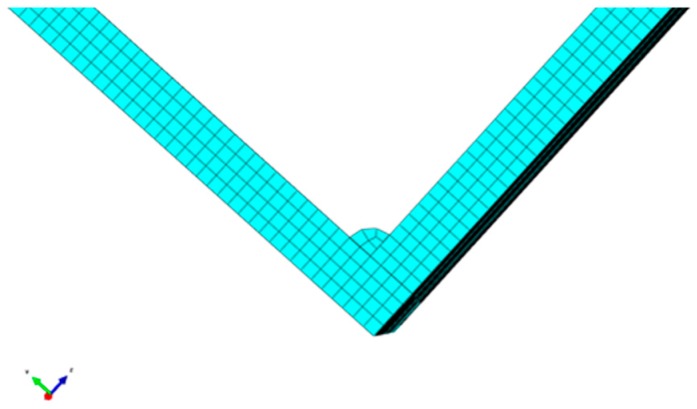
Grid diagram of welded steel plate.

**Figure 16 materials-12-01800-f016:**
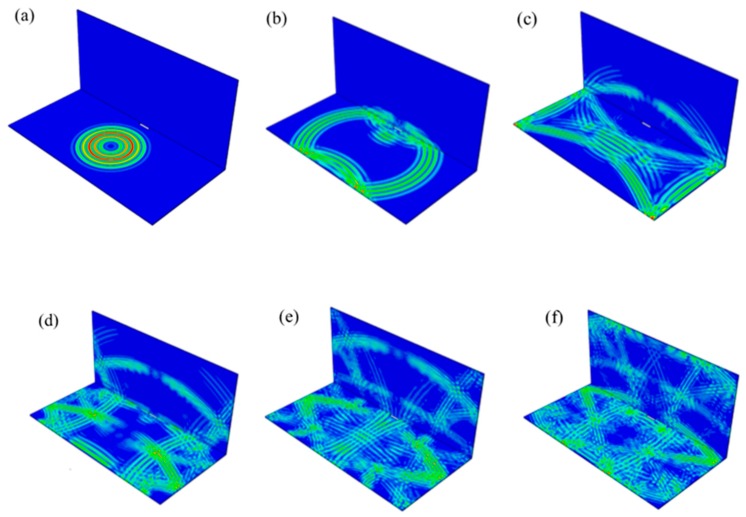
Propagation of Lamb waves in welded steel plates: (**a**) 2.5 × 10^−5^ s, (**b**) 3.0 × 10^−5^ s, (**c**) 3.5 × 10^−5^ s, (**d**) 4.0 × 10^−5^ s, (**e**) 4.5 × 10^−5^ s, (**f**) 5.0 × 10^−5^ s.

**Figure 17 materials-12-01800-f017:**
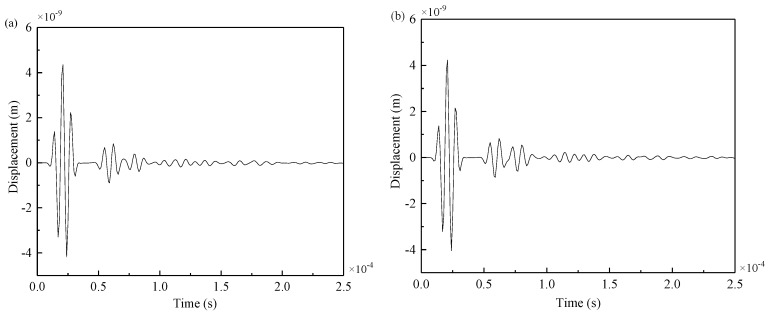
Signals from propagation path PZT5–PZT3: (**a**) the complete plate, (**b**) the damaged plate.

**Figure 18 materials-12-01800-f018:**
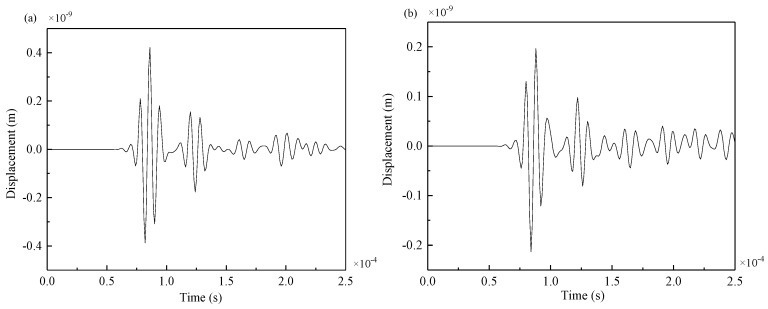
Signals from propagation path PZT5–PZT10: (**a**) the complete plate, (**b**) the damaged plate.

**Figure 19 materials-12-01800-f019:**
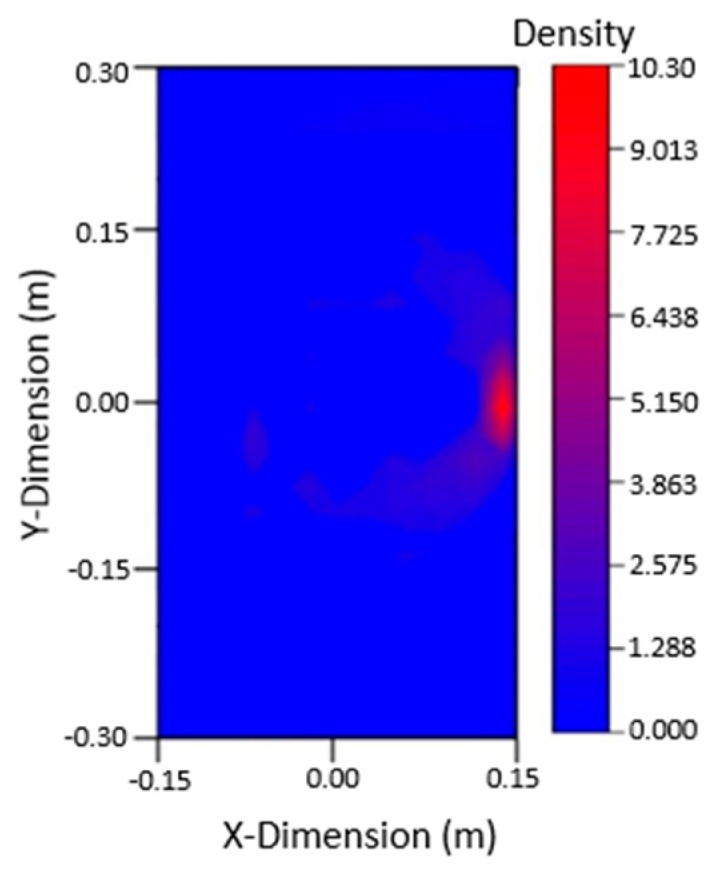
Imaging figure with 40 mm crack damage.

**Figure 20 materials-12-01800-f020:**
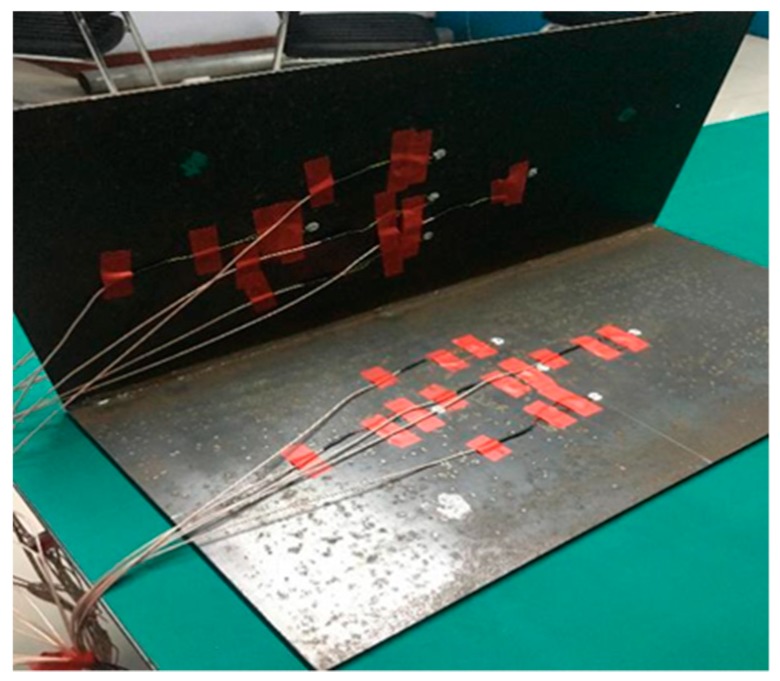
Experimental components and sensor layout.

**Figure 21 materials-12-01800-f021:**
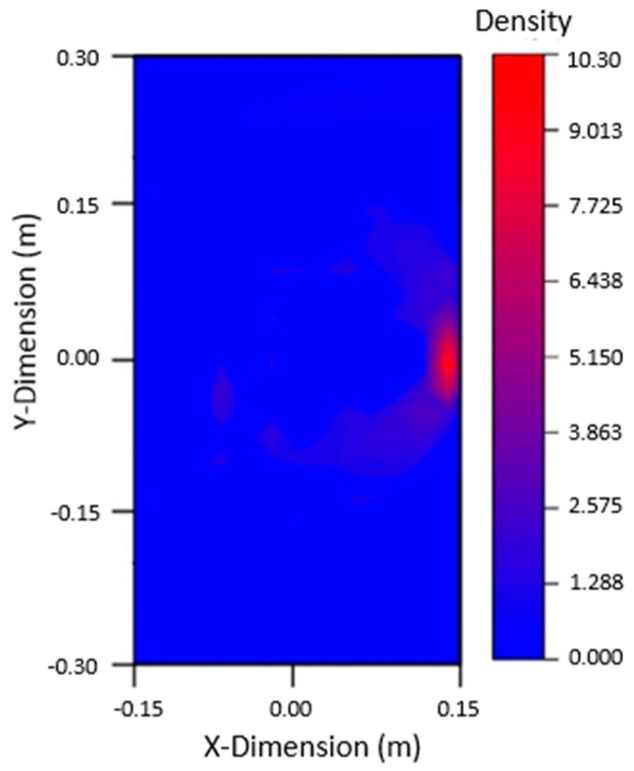
Imaging results of 40 mm crack damage.

**Figure 22 materials-12-01800-f022:**
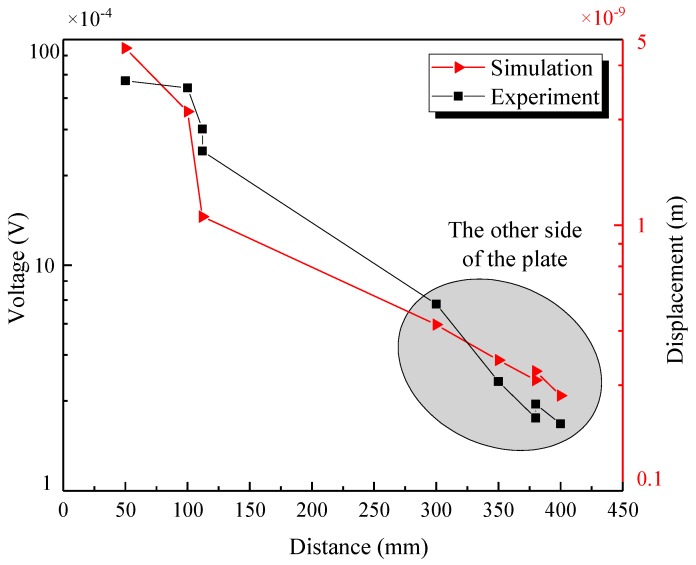
The amplitude and distance variation of the wave packet.

**Table 1 materials-12-01800-t001:** Material parameters for the Q235 steel.

E (GPa)	*ν*	*ρ* (kg/m^3^)	d (mm)	c_L_ (m/s)	c_T_ (m/s)
206	0.25	7800	4	5856	3130

**Table 2 materials-12-01800-t002:** Position coordinates of the sensors.

Transducer	Coordinate (mm)	Transducer	Coordinate (mm)	Transducer	Coordinate (mm)
PZT1	(0,160)	PZT4	(−160,0)	PZT7	(−80,−80)
PZT1’	(0,160)	PZT4’	(−160,0)	PZT7’	(−80,−80)
PZT2	(−80,80)	PZT5	(0,0)	PZT8	(80,−80)
PZT2’	(−80,80)	PZT5’	(0,0)	PZT8’	(80,−80)
PZT3	(80,80)	PZT6	(160,0)	PZT9	(0,−160)
PZT3’	(−80,80)	PZT6’	(160,0)	PZT9’	(0,−160)

**Table 3 materials-12-01800-t003:** Layout position of transducers.

Transducer (Left Side)	Coordinate (mm) (x,y)	Transducer (Left Side)	Coordinate (mm) (x,y)	Transducer (Right Side)	Coordinate (mm) (x’,y’)
PZT1	(0,50)	PZT3’	(0,0)	PZT6	(0,50)
PZT1’	(0,50)	PZT4	(100,0)	PZT7	(−100,0)
PZT2	(−100,0)	PZT4’	(100,0)	PZT8	(0,0)
PZT2’	(−100,0)	PZT5	(0,−50)	PZT9	(100,0)
PZT3	(0,0)	PZT5’	(0,−50)	PZT10	(0,−50)

**Table 4 materials-12-01800-t004:** The voltage of each receiving point in the experiment.

Propagation Path	Distance (mm)	Voltage (V) ×10^−3^	Propagation Path	Distance (mm)	Voltage (V) ×10^−3^
PZT1–PZT2	112	4.014	PZT1–PZT7	380	0.210
PZT1–PZT3	50	6.565	PZT1–PZT8	350	0.305
PZT1–PZT4	112	3.210	PZT1–PZT9	380	0.242
PZT1–PZT5	100	6.112	PZT1–PZT10	300	0.672
PZT1–PZT6	400	0.198			

**Table 5 materials-12-01800-t005:** The displacement of each receiving point in the numerical simulation.

Propagation Path	Distance (mm)	Displacement (mm)×10^−9^	Propagation Path	Distance (mm)	Displacement (mm) ×10^−9^
PZT1–PZT2	112	1.076	PZT1–PZT7	380	0.2608
PZT1–PZT3	50	4.642	PZT1–PZT8	350	0.3105
PZT1–PZT4	112	1.078	PZT1–PZT9	380	0.2819
PZT1–PZT5	100	2.679	PZT1–PZT10	300	0.4221
PZT1–PZT6	400	0.2283

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
