# Peer review of "Double Crack Damage Identification of Welded Steel Structure Based on LAMB WAVES of S0 Mode"

_materials, 2019, doi:10.3390/ma12111800_

Reviewer 1 Report

Paper deals with numerical and experimental investigation of Lamb wave propagation on steel plate and steel welded plate.

Major revisions are needed before considering it ready for publication. Reviewer suggests a proof reading of the paper. Moreover English must be significantly improved.

Abstract. Authors wrote: “…and damage to such structures will….” Rewrite, please.

Abstract. Authors wrote: “In this study, a Lamb wave is used as an acoustic emission signal to simulate the double crack damage of a steel plate”. Rewrite, please. This sentence does not make sense. Lamb waves are not used to simulate cracks, but they are used to detect cracks. Moreover, it does not make sense to speak of “a Lamb wave”, but of “Lamb waves”. Check the whole paper. For example, in introduction, page 1, authors wrote “Lamb wave has been an important means of nondestructive testing”.

In introduction, page 2, authors wrote “As noncontact nondestructive testing technique is viewed as viable solutions to meet the requirements like couplant-free, high testing efficiency, high reliability, real-time, visualized and environmental friendly, Ma and Zhou [12] applied this technology to the health monitoring of composite structure in aviation and aerospace industries” However, Lamb waves SHM systems are not used as noncontact inspection technique.

Introduce all acronyms, please. See IDT and PZT (page 2).

Table 1. Gpa should replace with GPa.

Table 1. Explain all parameters introduced in this table, please.

Separate measures from units of measure. For example, page 4: 800 mm x800 mmx4 mm, should be rewritten as 800 mm x 800 mm x 4 mm.

In order to improve the readability of the paper, it is necessary to indicate in figures 2 and 10 sensors labels.

Page 5, “The simulated frequency is 147 KHz and the minimum wavelength (  min) can be calculated as...”. Check ( ..min, please).

Equation 12, page 5. Authors wrote “The best principle of mesh generation is to transfer the minimum wavelength to 10 units. To satisfy this principle, the mesh length is calculated as follows”. This sentence should be referenced. According to reviewer’s point of view, a higher number of nodes per wavelength (NPW) improves the capability of the FE model in simulating the damage scattered waves. The minimum value of NPW is 10. Reviewer suggests considering the following reference: “A. De Luca, D. Perfetto, A. De Fenza, G. Petrone, F. Caputo, Guided waves in a composite winglet structure: Numerical and experimental investigations, Composite Structures 210 (2019) 96–108”. Specifically, this paper shows a mesh convergence analysis for the simulation of guided waves propagation in complex structures, characterized by strong curvatures.

More details about the developed numerical model should be added: number and type of elements, modelling of the sensors, modelling of the actuation signal, modelling of the adhesive between plate and sensors, boundary conditions. Specify the used FE code.

Figures 5, 13, 14 and 18. Amplitude units miss.

Page 9, “The central positions of the two cracks were (−245, 0) and (205, −60) respectively”. Specify the reference system.

Reviewer suggests showing the numerical-experimental comparison of some signals recorded at some sensors, in order to prove the reliability of the developed FE model.

Page 10. Give more details about the steel used for the welded plates.

Page 10, “On the left side, 10 PZTS are arranged symmetrically on both sides of the steel plates, and the serial numbers are pzt1-pzt5 and PZT1 '-pzt5'. Five PZTs were arranged on one side of the right steel plate, and the numbers are pzt6-10. The coordinates of left steel plate and right steel plate are shown in the figure. The specific coordinates of PZT paste on left side are listed in Table 3. The specific coordinates of PZT paste on right side are listed in Table 4”. Rewrite, please. Add sensors labels to Figure 10.

Specific the reference systems used in Table 3 and Table 4. Name differently the reference systems in Figure 10.

Table 5, specify the reference system.

Page 8. Specify the type of sensors.

Page 12, “Origin” was previously written in capital letters. Uniform it, please.  

Page 12, authors wrote: “Attenuation analysis of signal energy can be performed as follows: by comparing Figure 13 (a) with Figure 14 (a) and Figure 13 (b) with Figure 14 (b), it can be seen that signal energy decreases with the increase of distance regardless of whether there is damage in structural components”. Did authors consider damping factors in the model? Just in case, give more details. If authors did not model damping factors, they cannot state that the signal energy decreases due to the attenuation phenomenon. Signal energy decreases with the distance only if damping factors are introduced in the model. Consider the following reference in the paper: De Luca, A., Perfetto, D., De Fenza, A., Petrone, G., Caputo, F. A sensitivity analysis on the damage detection capability of a Lamb waves based SHM system for a composite winglet, 2018, Procedia Structural Integrity, 12, pp. 578-588.

Page 13. Replace 147 KHz with 147 kHz.

Page 14. FEM acronym has not been introduced before.

Author Response

Response to Reviewer 1 Comments

Dear Editors and Reviewer:

Thank you very much for your letter and reviewers’ comments concerning on our paper: “Double crack damage identification of welded steel structure based on Lamb waves of S0 mode”. Those comments are valuable and very helpful for us to improve our paper. We have studied the comments and have made plenty of corrections which we hope to meet with approval.

  The additions and modifications are highlighted in yellow in the paper. The main corrections in paper and responds to reviewers’ comments are listed as followings.

Point 1: Major revisions are needed before considering it ready for publication. Reviewer suggests a proof reading of the paper. Moreover English must be significantly improved.

Response 1: The English writing is revised thoroughly, we wish the modification can fulfill the requirements.

Point 2: Abstract. Authors wrote: “…and damage to such structures will….” Rewrite, please.

Response 2: We rewrote the sentence as :”The damage of steel structures will cause significant safety risks in a project, which can be seen in page 1.

Point 3: Abstract. Authors wrote: “In this study, a Lamb wave is used as an acoustic emission signal to simulate the double crack damage of a steel plate”. Rewrite, please. This sentence does not make sense. Lamb waves are not used to simulate cracks, but they are used to detect cracks. Moreover, it does not make sense to speak of “a Lamb wave”, but of “Lamb waves”. Check the whole paper. For example, in introduction, page 1, authors wrote “Lamb wave has been an important means of nondestructive testing”.

Response 3.1 : We rewrote the sentence as :” the propagation of Lamb waves in a steel plate with double cracks is simulated”, which can be seen in page 1.

Response 3.2: We changed all the “a Lame wave” to “Lamb waves”, all the modifications were highlighted in the paper.

Point 4: In introduction, page 2, authors wrote “As noncontact nondestructive testing technique is viewed as viable solutions to meet the requirements like couplant-free, high testing efficiency, high reliability, real-time, visualized and environmental friendly, Ma and Zhou [12] applied this technology to the health monitoring of composite structure in aviation and aerospace industries” However, Lamb waves SHM systems are not used as noncontact inspection technique.

Response 4: We rewrote this part of the introduction as :” As nondestructive testing technique is viewed as viable solutions to meet the requirements like couplant-free, high testing efficiency… and proposed a damage imaging method that accommodates for multipath scattering Lamb wave signals.” And we replaced a reference article which is more consistent with the content, which can be seen in page 2.

Point 5: Introduce all acronyms, please. See IDT and PZT (page 2).

Response 5: We introduced the acronyms of IDT and PZT in the sentence :” Takpara et al. [16] validated the optimization of parameters for the InterDigital Transducer (IDT) sensor with Piezoelectric ceramics Transducer (PZT)”, which can be seen in page 2.

Point 6: Table 1. Gpa should replace with GPa.

Response 6: We replaced the “Gpa” with “GPa”, which can be seen in page 4.

Point 7: Table 1. Explain all parameters introduced in this table, please.

Response 7: We added the explanations of the parameters : “Where E is the elastic modulus of the steel plate, v is Poisson’s ratio, r is the density of the material, d is the thickness of the steel plate, cL is the velocity of longitudinal wave, cT is the velocity of transverse wave”. The additions can be seen in page 3.

Point 8: Separate measures from units of measure. For example, page 4: 800 mm x800 mmx4 mm, should be rewritten as 800 mm x 800 mm x 4 mm.

Response 8: We separated measures from units of measure:

1. “800 mm ´ 800 mm ´ 4 mm” in page 4;

2. “2.5 ´ 10-5 s to 5.0 ´ 10-5 s”, “5 ´ 10-6 s” and “(a) 2.5 ´ 10-5 s, (b) 3.0 ´ 10-5 s, (c) 3.5 ´ 10-5 s, (d) 4.0 ´ 10-5 s, (e) 4.5 ´ 10-5 s, (f) 5.0 ´ 10-5 s” in page 8.

3. “600 mm × 300 mm × 4 mm” in page 11.

4. “2.5 ´ 10-5 s to 5.0 ´ 10-5 s”, “5 ´ 10-6 s” and “(a) 2.5 ´ 10-5 s, (b) 3.0 ´ 10-5 s, (c) 3.5 ´ 10-5 s, (d) 4.0 ´ 10-5 s, (e) 4.5 ´ 10-5 s, (f) 5.0 ´ 10-5 s” in page 12.

Point 9: In order to improve the readability of the paper, it is necessary to indicate in figures 2 and 10 sensors labels.

Response 9: The Fig.2 and Fig.10 have adjusted to Fig. 3 and Fig.14, and we added the sensors labels in Fig.3 and Fig.14, which can be seen in page 5 and page 11.

Point 10: Page 5, “The simulated frequency is 147 KHz and the minimum wavelength (  min) can be calculated as...”. Check ( ..min, please).

Response 10: We rewrote the sentence as “The simulated frequency is 147 kHz and the minimum wavelengthlmincan be calculated as:”, the modification was highlighted in section 3.3, which can be seen in page 6.

Point 11: Equation 12, page 5. Authors wrote “The best principle of mesh generation is to transfer the minimum wavelength to 10 units. To satisfy this principle, the mesh length is calculated as follows”. This sentence should be referenced. According to reviewer’s point of view, a higher number of nodes per wavelength (NPW) improves the capability of the FE model in simulating the damage scattered waves. The minimum value of NPW is 10. Reviewer suggests considering the following reference: “A. De Luca, D. Perfetto, A. De Fenza, G. Petrone, F. Caputo, Guided waves in a composite winglet structure: Numerical and experimental investigations, Composite Structures 210 (2019) 96–108”. Specifically, this paper shows a mesh convergence analysis for the simulation of guided waves propagation in complex structures, characterized by strong curvatures.

Response 11: The sentence “The best principle of mesh generation is to transfer the minimum wavelength to 10 units [17].” was referenced. The reference is “A. De Luca, D. Perfetto, A. De Fenza, G. Petrone, F. Caputo, Guided waves in a composite winglet structure: Numerical and experimental investigations, Composite Structures 210 (2019) 96–108”. The modification can be seen in section 3.3, page 7.

Point 12: More details about the developed numerical model should be added: number and type of elements, modelling of the sensors, modelling of the actuation signal, modelling of the adhesive between plate and sensors, boundary conditions. Specify the used FE code.

Response 12.1: number and type of elements: we added the details of the elements:

1. “As a result, the plate has been modelled with 2560596 three-dimensional eight-node linear brick elements (C3D8R).” The modification can be seen in section 3.3 , page 7.

2.“As a result, the left plate has been modelled with 713376 three-dimensional eight-node linear brick elements (C3D8R)…and the weld joint has been modelled with 3000 C3D8R.”  The modification can be seen in section 4.1, page 11.

Response 12.2: modelling of the sensors, modelling of the adhesive between plate and sensors, boundary conditions:  We added: “In this model, the excitation is demonstrated as a tangentially concentrated load, the load is a time period dependent dynamic load…at the place of piezoceramics as shown in Fig.6.” and Fig.6 to explain these details. The additions can be seen in section 3.2, page 6.

Response 12.3: Specify the used FE code: we added the details: “The numerical modelling has been performed by using ABAQUSâ ver. 6.14 finite element code”, which can be seen in section 3.1, page 4.

Point 13: Figures 5, 13, 14 and 18. Amplitude units miss.

Response: The Fig.5, 13, 14, 18 have adjusted to Fig. 9, 17, 18, 22, and We added the Amplitude units of these figures , which can be seen in page 9, 13, 16.

Point 14: “The central positions of the two cracks were (−245, 0) and (205, −60) respectively”. Specify the reference system.

Response: We added the reference system in Fig.12. Which can be seen in section 3.6, page 10.

Point 15: Reviewer suggests showing the numerical-experimental comparison of some signals recorded at some sensors, in order to prove the reliability of the developed FE model.

Response: Thanks for your suggestion, we added the comparison between numerical simulation and experimental results in section 4.3.2: “In the numerical simulation,…the logarithmic coordinate is applied in the ordinate.”, Table 4, Table5 and Fig.22. The details can be seen in page 15-16.

Point 16: Page 10. Give more details about the steel used for the welded plates.

Response 16: We added more details about the welded plates: “The parameters of the material can be seen in Table 1 in section 2.2”, “The specific coordinates of PZTs are listed in Table 3.”, “1 mm is chosen as the mesh size in the simulation model… with 3000 C3D8R.” and Table 3. The addition can be seen in section 4.1,  page 11-12.

Point 17: Page 10, “On the left side, 10 PZTS are arranged symmetrically on both sides of the steel plates, and the serial numbers are pzt1-pzt5 and PZT1 '-pzt5'. Five PZTs were arranged on one side of the right steel plate, and the numbers are pzt6-10. The coordinates of left steel plate and right steel plate are shown in the figure. The specific coordinates of PZT paste on left side are listed in Table 3. The specific coordinates of PZT paste on right side are listed in Table 4”. Rewrite, please. Add sensors labels to Figure 10.

Response 17: We rewrote this part: “On the left side, 10 PZTS are arranged symmetrically on both sides of the steel plates…as shown in the Fig.14”, which can be seen in section 4.1, page 11. And we added sensors labels to Fig.14 (original Fig.10).

Point 18: Specific the reference systems used in Table 3 and Table 4. Name differently the reference systems in Figure 10.

Response 18: We added the reference systems of left plate and right plate in the Fig.14 (original Fig.10), and named them differently, the reference systems were specified in Table 3. These details can be seen in section 4.1, page 11-12.

Point 19: Table 5, specify the reference system.

Response 19: We modified the Table 4 (original Table 5), and explained the parameters in the table, which can be seen in section 4.3.2, page 15.

Point 20: Page 8. Specify the type of sensors.

Response 20: The meaning of the PZT can be seen in page 2: “Takpara et al. [16] … with Piezoelectric ceramics Transducer (PZT).

The type of the sensors was introduced as “The type of PZT is YT-5L with a diameter of 10mm and the thickness of 1 mm.”, which can be seen in section 3.6, page 10.

Point 21: “Origin” was previously written in capital letters. Uniform it, please. 

Response 21: We uniformed the written to “ORIGIN”,  which were highlighted in page 9 and 13.

Point 22: authors wrote: “Attenuation analysis of signal energy can be performed as follows: by comparing Figure 13 (a) with Figure 14 (a) and Figure 13 (b) with Figure 14 (b), it can be seen that signal energy decreases with the increase of distance regardless of whether there is damage in structural components”. Did authors consider damping factors in the model? Just in case, give more details. If authors did not model damping factors, they cannot state that the signal energy decreases due to the attenuation phenomenon. Signal energy decreases with the distance only if damping factors are introduced in the model. Consider the following reference in the paper: De Luca, A., Perfetto, D., De Fenza, A., Petrone, G., Caputo, F. A sensitivity analysis on the damage detection capability of a Lamb waves based SHM system for a composite winglet, 2018, Procedia Structural Integrity, 12, pp. 578-588.

Response 22: Thanks for your suggestion and reference. In the previous study, the damping effect of the surrounding medium on particle vibration was considered in the numerical simulation. Now, based on your suggestions and our research before, the influence of material damping factor was considered in the simulation. We gave the details of the damping factor in section 4.1: “The damping factor of the material is set at 0.005”, which can be seen in Fig.11. We recalculated the numerical simulation of this section and corrected the results of the numerical simulation, as shown in Fig.17, 18, which can be seen in 13.

Point 23: Page 13. Replace 147 KHz with 147 kHz.

Response 23: The “KHz” were replaced with “kHz”, which were highlighted in page 6 and page 15.

Point 24: FEM acronym has not been introduced before.

Response 24: “FEM” was replaced with “finite element method” in the conclusion, which can be seen in conclusion 1, page 16.

In conclusion, the authors are invited to perform a deeper scrutinizing before to submit a revised manuscript. These are our responses to reviewers. Besides we adjust main frame of paper and re-written most of contents.

Best regards

Jian He

Reviewer 2 Report

Comments to the Author(s):

1.     Please explain the mode selection in detail.

2.     Basic theory of Lamb wave: The authors must include appropriate citations against the formations.

3.     Please brief  the figure captions (e.g. Fig.1).

4.     Eq. 9-12: please include the citations.

5.     Fig. 5, 13-14: I suggest the authors to start the X-axis values from zero.

6.     The Authors should maintain the sentence case for all the Fig. captions (e.g. Fig. 4 is a mix of upper & Lower case). Please check thoroughly & modify accordingly.

7.     Brief further & only write the specific contributions in the conclusions.

8.     I suggest the Author (s) to check the paper for a thorough English correction, break down the long scentences and modify accordingly.

9.     The literature review should be improved by including more relevant and recent citations. The authors are recommended to include the following citations in the first paragraph of the Introduction (i.e. Line No. 27-32) :

i.               Journal: “Damage-induced acoustic emission source monitoring in a honeycomb sandwich composite structure”

ii.             Journal: “Acoustic emission based damage localization in composites structures using Bayesian identification”

iii.            Journal: “Damage-induced acoustic emission source identification in an advanced sandwich composite structure”

iv.            Book: “Structural Health Monitoring of Advanced Composites Using Guided Waves”.

10.  The methodology need to be explain with more details and the Author(s) must try to improve the paper with some more interesting results.

Author Response

Response to Reviewer 2 Comments

 Dear Editors and Reviewer:

  Thank you very much for your letter and reviewers’ comments concerning on our paper: “Double crack damage identification of welded steel structure based on Lamb waves of S0 mode”. Those comments are valuable and very helpful for us to improve our paper. We have studied the comments and have made plenty of corrections which we hope to meet with approval.

  The additions and modifications are highlighted in blue in the paper. The main corrections in paper and responds to reviewers’ comments are listed as followings.

Point 1: Please explain the mode selection in detail.

Response 1: In order to explain the mode selection of the Lamb waves thoroughly, we added some theories “which are as shown in Fig.4…. conducive to wave packet separation.”, and Fig.4, Fig.5 to explain the reasons for mode selection, which can be seen in the section 3.2, page 5-6.

Point 2:   Basic theory of Lamb wave: The authors must include appropriate citations against the formations.

Response 2: On the basis of ensuring the accuracy of the article, we added some explanations to the formula on page3: “the formula for calculating group velocity can be obtained as follows”.

Point 3:   Please brief the figure captions (e.g. Fig.1).

Response 3: We adjust all the figure captions from Figure.A to Fig.A.

Point 4:   Eq. 9-12: please include the citations.

Response 4: Some appropriate citations are added to the Eq. 9-12 from page5 to 7: “The expression of”“The calculation process”“The simulated frequency…”.

Point 5:   Fig. 5, 13-14: I suggest the authors to start the X-axis values from zero.

Response 5: The Fig.5, 13-14 have adjusted to Fig. 9 in page 9, Fig. 17-18 in page 13, the X-axis values of these figures are all adjusted to start from zero.

Point 6:   The Authors should maintain the sentence case for all the Fig. captions (e.g. Fig. 4 is a mix of upper & Lower case). Please check thoroughly & modify accordingly.

Response 6: We check all the figure captions and modify these issues carefully.

Point 7:   Brief further & only write the specific contributions in the conclusions.

Response 7: After adjusting the content of the article, we rewrite the conclusions as brief as possible in page 16. Only summarize the main conclusions of this paper.

Point 8:   I suggest the Author (s) to check the paper for a thorough English correction, break down the long sentences and modify accordingly.

Response 8: The English writing is revised thoroughly, we wish the modification can fulfill the requirements.

Point 9:   The literature review should be improved by including more relevant and recent citations. The authors are recommended to include the following citations in the first paragraph of the Introduction (i.e. Line No. 27-32)

i.               Journal: “Damage-induced acoustic emission source monitoring in a honeycomb sandwich composite structure”

ii.             Journal: “Acoustic emission based damage localization in composites structures using Bayesian identification”

iii.            Journal: “Damage-induced acoustic emission source identification in an advanced sandwich composite structure”

iv.            Book: “Structural Health Monitoring of Advanced Composites Using Guided Waves”.

Response 9: Considering the relevance to the content of the article, three of these references have been added, and introduction part was also re-written to make it more clearly and more relevant in page1:“For example…. composite structures was proposed”. References related in the section have been quoted in corresponding places.

Point 10:   The methodology need to be explain with more details and the Author(s) must try to improve the paper with some more interesting results.

Response 10: We added many details of the methodology:

  1. “Unilateral excitation is not conducive to signal extraction because it generates two different modes of excitation wave…The wave group of S0 mode is fast, that is conducive to wave packet separation.”, and Fig.4 in section 3.2, page 5.

  2. “In this model,…and simulates the establishment of cylindrical coordinates at the excitation load at the place of piezoceramics as shown in Fig.6”, and Fig.6 in section 3.2, page 6.

  3. “As a result, the plate has been modelled with 2560596 three-dimensional eight-node linear brick elements (C3D8R).” in section 3.3, page 7.

  4. “1 mm is chosen as the mesh size in the simulation model…and the weld joint has been modelled with 3000 C3D8R.” in section 4.1, page 11.

And we Revised the content of the conclusion.

 In conclusion, the authors are invited to perform a deeper scrutinizing before to submit a revised manuscript. These are our responses to reviewers. Besides we adjust main frame of paper and re-written most of contents.

Best regards

Jian He

Round  2

Reviewer 1 Report

Authors have properly addressed all raised comments and questions. However, English have to be still improved before considering it ready for publication. Moreover, some formal changes are still needed. For example, reviewer found some typing mistakes.

Author Response

Dear Editors and Reviewer:

Thank you very much for your letter and reviewers’ comments concerning on our paper: “Double crack damage identification of welded steel structure based on Lamb waves of S0 mode”.

Point 1: Authors have properly addressed all raised comments and questions. However, English have to be still improved before considering it ready for publication. Moreover, some formal changes are still needed. For example, reviewer found some typing mistakes.

Response 1:The English writing has been improved thoroughly, and the typing mistakes have been corrected, the additions and modifications are highlighted in yellow in the paper. We wish the modification can meet the requirements. 

Best regards

Jian He

Reviewer 2 Report

I appreciate the Authors efforts towards the improvement of the paper. I recommend the paper for publication.

Author Response

Dear Editors and Reviewer:

  Thank you very much for your letter and reviewers’ comments concerning on our paper: “Double crack damage identification of welded steel structure based on Lamb waves of S0 mode”.

  We are very appreciate for your comments and time for this paper!

Best regards

Jian He